# Structural basis for higher-order DNA binding by a bacterial transcriptional regulator

**Frederik Oskar Graversgaard Henriksen**, **Lan Bich Van, Ditlev Egeskov Brodersen**, **Ragnhild Bager Skjerning**\*

Department of Molecular Biology and Genetics, Aarhus University, Aarhus, Denmark

\* raba@mbg.au.dk

## Abstract

Transcriptional regulation by binding of transcription factors to palindromic sequences in promoter regions is a fundamental process in bacteria. Some transcription factors have multiple dimeric DNA-binding domains, in principle enabling interaction with higher-order DNA structures; however, mechanistic and structural insights into this phenomenon remain limited. The *Pseudomonas putida* toxin-antitoxin (TA) system Xre-RES has an unusual 4:2 stoichiometry including two potential DNA-binding sites, compatible with a complex mechanism of transcriptional autoregulation. Here, we show that the Xre-RES complex interacts specifically with a palindromic DNA repeat in the promoter in a 1:1 molar ratio, leading to transcriptional repression. We determine the 2.7 Å crystal structure of the protein-DNA complex, revealing an unexpected asymmetry in the interaction and suggesting the presence of a secondary binding site, which is supported by structural prediction of the binding to the intact promoter region. Additionally, we show that the antitoxin can be partially dislodged from the Xre-RES complex, resulting in Xre monomers and a 2:2 Xre-RES complex, neither of which repress transcription. These findings highlight a dynamic, concentration-dependent model of transcriptional autoregulation, in which the Xre-RES complex transitions between a non-binding (2:2) and a DNA-binding (4:2) form.

## Author summary

Bacteria regulate their gene expression to respond to environmental stress, to evade antibiotics, and to maintain population stability. In this study, we investigate how the *xre-res* toxin-antitoxin system from *Pseudomonas putida* controls its own expression. Using a combination of microbiology, structural biology, biophysical assays, and computational modelling, we discover how the Xre-RES protein complex represses its own transcription through direct binding to a specific DNA element in the promoter region. We show structurally that the Xre-RES complex adopts a unique 4:2 stoichiometry and binds DNA in an unusual

**Data availability statement:** All relevant data are within the manuscript and its Supporting Information files.

**Funding:** This study was funded by the Independent Research Fund Denmark, grant no. 0135-00072B and by the Novo Nordisk Foundation, grants NNF18OC0030646 and NNF22OC0079855 to D.E.B. The funders had no role in study design, data collection and analysis, decision to publish, or preparation of the manuscript.

**Competing interests:** The authors have declared that no competing interests exist.

asymmetrical manner. Moreover, the complex was found to shift between two different forms: one that binds DNA and represses transcription, and one that does not. We further demonstrate that this shift is dynamic and depends on the relative concentration of Xre antitoxin. Our findings provide new insight into how bacteria fine-tune gene expression and offers a model of transcriptional control based on protein stoichiometry and structural asymmetry.

## Introduction

Regulation of transcription is a fundamental mechanism in bacteria that allows adaptation to constantly changing environments, and RNA polymerase (RNAP) plays a central part in this. The core RNAP enzyme has a $\beta\beta'\alpha_2$ subunit composition [1] but in order to form a holoenzyme and initiate transcription at a particular promoter, the core enzyme needs to interact with a σ subunit [2,3]. The σ subunit ensures promoter recognition and positioning of the RNAP holoenzyme at the promoter and unwinding of the DNA duplex near the transcriptional start site (TSS). Transcriptional activity can be modulated by repressor proteins that bind directly to a segment of the promoter (the operator) and inhibits transcription [4]. Repression can occur by one of three mechanisms: steric hindrance of RNAP binding to the promoter, interference with post-recruitment steps in transcriptional initiation such as the assembly of the RNAP holoenzyme, or DNA looping upon repressor binding [1]. A common feature of many repressor proteins is the presence of a DNA-binding helix-turn-helix (HTH) domain [5]. The generic HTH domain is a simple fold comprised of three core helices of which the second and third helix are connected by a sharp turn and makes up the HTH motif [5,6]. The third helix is also called the 'recognition helix' and usually recognizes a symmetrical sequence element in the DNA by inserting directly into the major groove [7]. The binding efficiency of a repressor protein to an operator site depends on its binding affinity, which again is determined by base-specific readouts and intrinsic DNA shape, however, in many cases, the mechanism by with repressor proteins recognise DNA *in vivo* remains unclear [8–10].

Due to the symmetrical nature of DNA, most HTH-containing protein repressors are functional as dimers, which sometimes depend on even higher-order structures and additional proteins. Thus, HTH domains often bind palindromic DNA sequences separated by approximately a single helical turn, allowing for the recognition helices to insert into adjacent major grooves [7]. In the widespread *xre-res* system [11], the encoded Xre (Xenobiotic response element) protein contains a putative HTH DNA-binding domain that is stabilised in a higher-order structure through interaction with the associated RES-domain protein [12–14]. The two proteins are expressed from two-gene operons that have been classified as toxin-antitoxin (TA) systems, which are widely distributed cellular regulators in bacteria that encode a protein toxin that inhibits cell growth along with a cognate protein antitoxin that neutralises it [15]. The RES toxin has an ADP-ribosyltransferase fold and expression of an orthologue from the insect pathogen, *Photorhabdus luminescens*, has been shown to result

in efficient depletion of intracellular NAD$^+$ in *E. coli*, suggesting that the RES toxin is a NAD$^+$ glycohydrolase (NADase). Similar results were obtained for the RES-domain toxins MbcT from *Mycobacterium tuberculosis* [12], SlvT from *P. putida* S12 [16], PA14_51010 from *P. aeruginosa* PA14 [17], and VPA0770 from *Vibrio parahaemolyticus* [18], while PrsT from *Sinorhizobium meliloti* toxin is structurally similar but appears to function as an ADP ribosyltransferase (mART) targeting and inactivating phosphoribosyl pyrophosphate synthetase (Prs) [13]. The role of RES domain-containing TA systems in bacterial cell biology is still largely unknown, but a recent study showed that the *P. aeruginosa* orthologue, PA14_51010 (NatT), mediates NAD$^+$ depletion leading to drug-tolerant dormant cell populations *in vitro*, suggesting that the accumulation of NatT-activating mutations observed during *P. aeruginosa* infections in human cystic fibrosis patients increases antibiotic tolerance [19].

The crystal structure of a Xre-RES complex from *Pseudomonas putida* revealed an unusual 4:2 TA stoichiometry in which each of the RES domains of a central toxin dimer interacts with a dimer of Xre antitoxins, thus generating two complete HTH domains [14]. The HTH domain of this Xre antitoxin (Xre$^{Pp}$) is structurally similar to the bacteriophage lambda Cro repressor [20] and could thus potentially play a part in transcriptional regulation and toxin control through binding of the antitoxin to the TA promoter region, as observed for many TA systems [21]. For some of these, the presence of the toxin only stabilises the interaction with DNA up to a certain level after which a higher stoichiometry of toxin molecules leads to de-repression by a mechanism termed *conditional cooperativity* [22–24]. However, other modes of autoregulation exist, including recruitment of external regulators, three-component TA systems in which the DNA binding domain is present in a third protein separate from the antitoxin, and TA modules with reversed gene order and multiple promoter sites [25]. The observation that the *P. putida* Xre-RES complex contains two complete DNA-binding domains each composed by two HTH domains suggests the system may be regulated in a novel way. However, the organisation of the promoter region and the mechanism of DNA binding and transcriptional regulation are currently unknown.

In this paper, we dissect the mechanism of transcriptional autoregulation of the *P. putida xre-res* system (*xre-res*$^{Pp}$). Our findings reveal that a Xre–RES$^{Pp}$ complex with 4:2 stoichiometry binds in a 1:1 ratio to an imperfect inverted repeat region in the promoter region and effectively represses transcription. We determine a 2.7 Å crystal structure of the Xre-RES$^{Pp}$ complex bound to the imperfect inverted repeat, showing that the interaction is unusually asymmetric. Moreover, interactions to one end of a symmetry-related DNA-duplex suggests a second binding site in the operator region, which is supported by structure prediction of the interaction to the intact promoter DNA sequence. Finally, we show that the isolated Xre$^{Pp}$ antitoxin is a monomer that only weakly represses the promoter. Surprisingly, individual Xre$^{Pp}$ monomers can dislodge from the 4:2 complex, resulting in a 2:2 complex incapable of binding DNA. Together, our data provides a model for how the Xre-RES$^{Pp}$ complex autoregulates transcription through a balance between binding and non-binding forms, driven by protein stoichiometry and asymmetry, as well as general insights into interaction of transcription factors with higher-order DNA in bacteria.

## Results

### The *P. putida xre-res* locus is transcriptionally auto-regulated

The presence of a DNA-binding HTH domain in Xre$^{Pp}$ suggested that the antitoxin and/or the Xre-RES TA complex could be involved in regulating transcription of the *P. putida xre-res* operon via direct promoter binding. To test this hypothesis, we analysed the *xre-res*$^{Pp}$ promoter region (P$_{XR}$) and identified three inverted repeats and one directional repeat, named Sequence 1 to 4 (S1-S4) (Figs 1A and S1A). To investigate the promoter activity *in vivo*, we constructed a low-copy GFP reporter plasmid (pGH254Kgfp) containing the intact P$_{XR}$ promoter region, including the 5' end (30 bp) of *xre*$^{Pp}$, transcriptionally fused to *gfpmut2* encoding a Green Fluorescent Protein (GFP) variant (S65A/V68L/S72A) with enhanced fluorescence emission and more efficient folding at 37°C as compared to the wild type GFP [26]. The resulting construct, as well as a plasmid expressing *xre-res*$^{Pp}$ under arabinose-inducible control, was introduced into *E. coli* MG1655 by transformation, and the GFP signal monitored throughout growth in the presence or absence of arabinose (Fig 1B). The results

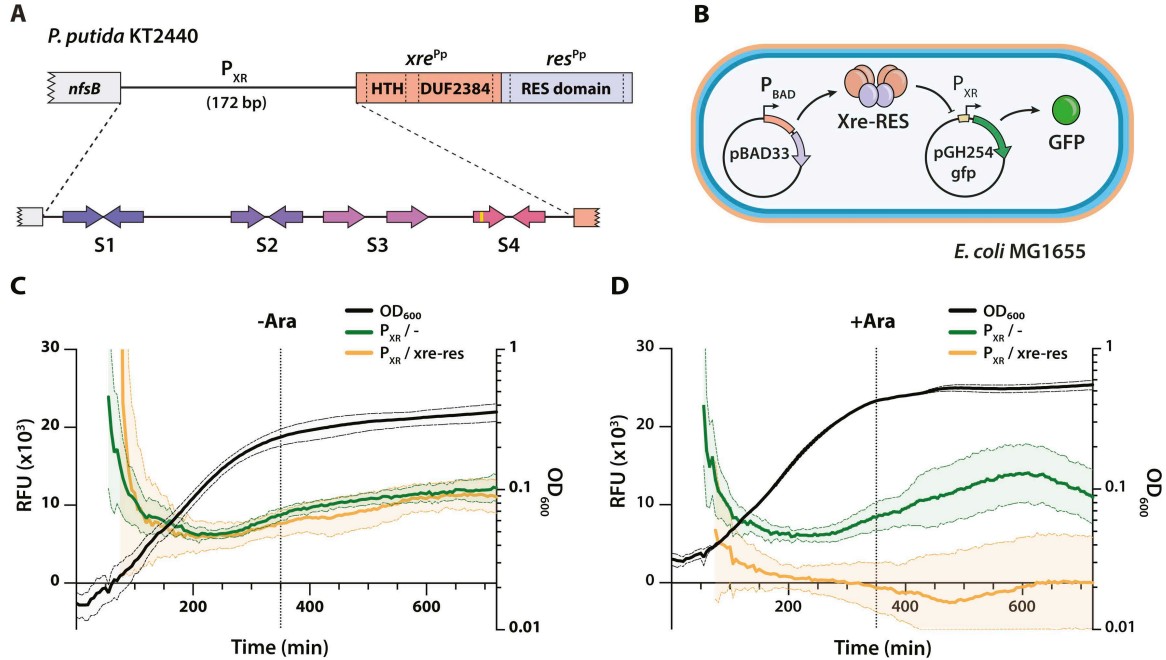

**Fig 1. Organisation, activity, and regulation of the *xre-res*^Pp operon.** (A) Schematic representation the *P. putida* KT2440 *xre-res* locus (*xre-res*^Pp) including the 172-bp promoter region (P$_{XR}$) and the upstream gene (*nfsB*). Top, the *xre*^Pp gene (orange) encodes a protein antitoxin with an N-terminal helix-turn-helix (HTH) domain and a C-terminal domain of unknown function (DUF2384), known to interact with the RES toxin, while *res*^Pp (blue) encodes a NADase toxin with a RES domain. Below, close-up of the P$_{XR}$ promoter region showing the three inverted repeats (S1, S2, and S4) and one directional repeat (S3), of which S4 is imperfect and contains one mismatch (yellow box, see also S1A Fig). (B) Overview of the GFP-based assay to measure regulation of P$_{XR}$ in *E. coli* MG1655 *in vivo*. The pBAD33 vector allows for arabinose induction of the *xre-res*^Pp genes while P$_{XR}$ coupled to GFP allows for read-out of the transcriptional activity of the promoter. (C) and (D) Promoter activity assays in *E. coli* MG1655. Activity is measured as GFP signal in relative fluorescence units (RFU) during growth without ((C), -Ara) or with ((D), +Ara) arabinose using the promoter reporter pGH254Kgfp::P$_{XR}$ (P$_{XR}$), as well as empty pBAD33 (-, green curves) or pBAD33::*xre-res*^Pp vector (*xre-res*, orange curves). All curves represent mean-of-mean values (line, *n*=3) with standard error of mean (SEM) shown as a shadow. The dotted vertical line indicates the 350 min time point.

show that the P$_{XR}$ promoter is active in *E. coli* during exponential growth (Fig 1C). Moreover, induction of *xre-res*^Pp in this background results in complete absence of GFP signal (Fig 1D, orange curve) compared to the strain containing an empty pBAD33 vector (Fig 1D, green curve). Together, this supports that *xre-res*^Pp negatively regulates expression from P$_{XR}$.

## Expression of *xre-res*^Pp represses promoter activity via an imperfect repeat upstream of *xre*^Pp

To determine the minimal promoter requirements for active transcription from the *xre-res*^Pp locus, we repeated the assay using variants of P$_{XR}$ where one or more of the four identified repeats (S1-S4) were systematically removed (Fig 2A and S1 Table). To simplify the comparison and since the P$_{XR}$ promoter was found to be most active during late exponential growth, the GFP fluorescence level measured at t=350 min was used for this analysis (complete curves are included in S1 and S2 Figs). Removal of S1 (ΔS1) did neither affect promoter activity nor repression by *xre-res*^Pp, while removal of both S1 and S2 (ΔS1-S2), resulted in significantly increased promoter activity that can still be fully repressed by *xre-res*^Pp (Fig 2B). In contrast, removal of either S1 through S3 (ΔS1-S3), S3 and S4 (ΔS3–4), or just S4 (ΔS4), completely abolished transcription (Fig 2B). Interestingly, removal of only the central palindromic part of the imperfect inverted repeat in S4 (Δre-peat, Fig 2C and S1 Table) did not affect promoter activity while expression of *xre-res*^Pp only weakly (and non-significantly) repressed transcription of this variant (Fig 2D). Together, this suggests that transcriptional control by *xre-res*^Pp depends on the imperfect inverted repeat in the S4 region while transcription does not. To investigate the importance of the imperfect

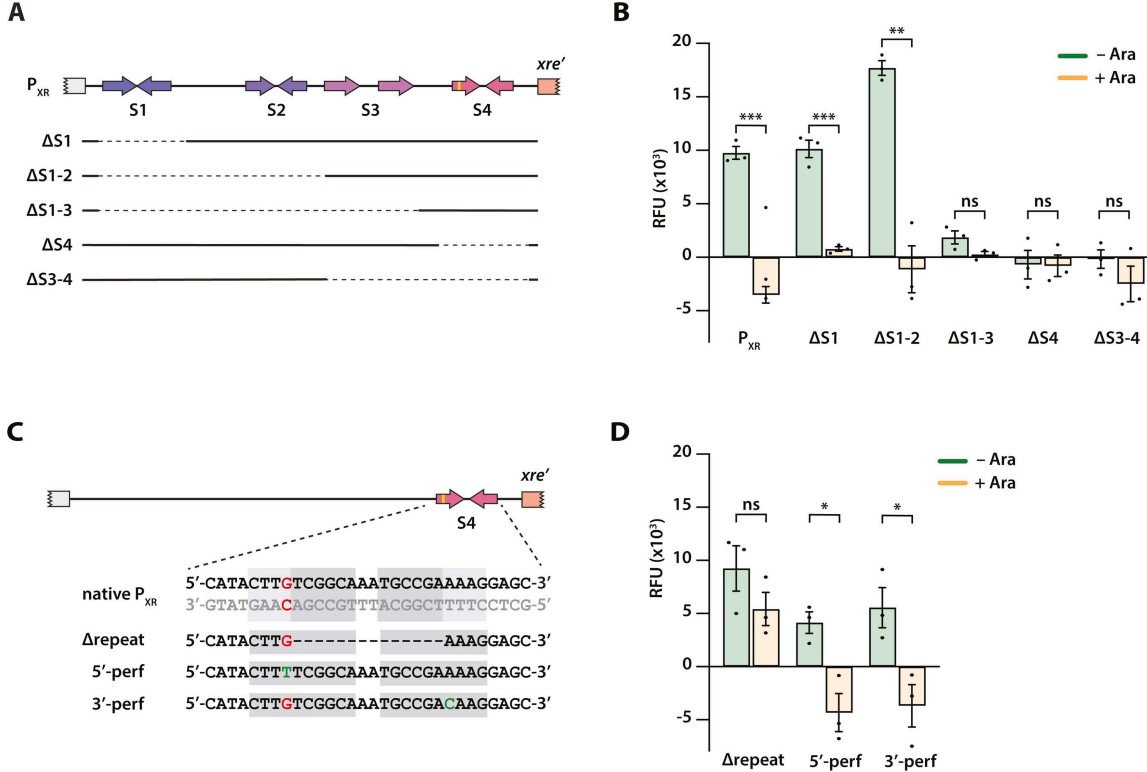

**Fig 2. Determination of the minimal promoter requirements for transcription and regulation of P$_{XR}$.** (A) Detailed overview of the P$_{XR}$ promoter (top) and schematic representation of promoter variants (below). Full lines indicate the included parts while dotted lines represent regions that have been removed. Complete growth curves and details of the constructs are found in S1 Fig and S1 Table. (B) Promoter activity assays in *E. coli* MG1655. The reporter plasmid pGH254Kgfp encoding promoter variations was combined with pBAD33::*xre-res*$^{Pp}$. For each variant, the GFP signal (RFU) measured at t = 350 min from non-induced cultures (-Ara, green bars) is compared to that from arabinose-induced cultures (+Ara, orange bars). Mean RFU values extracted at the same time point from three independent experiments were treated as biological replicates (dots, $n = 3$) and error bars show SEM. Complete growth curves are included in S1 Fig. Statistical analysis was performed using Welch's t-test (with equal variance). *ns*, not significant; **-, $p < 0.01$; ***, $p < 0.001$. (C) Schematic representation of the repeat variants of the imperfect inverted repeat in S4 of the P$_{XR}$ promoter region. The DNA sequence of the imperfect inverted repeat is highlighted (light grey), including the mismatch in the first repeat (red), the perfect part of the repeat (dark grey) and the alterations (green). The mismatch was removed by a substitution either in the first (5'-perf) or second (3'-perf) half of the repeat. (D) As b, but using promoter variations from c. *ns*, not significant; *, $p < 0.05$.

part of the inverted repeat, we created two variants with perfect repeats by either introducing G->T on the coding strand in the first half of the repeat (5'-perf) or A->C in the second half of the repeat (3'-perf) (Fig 2C and S1 Table). Both substitutions decreased promoter activity as compared to wild type but did not affect the ability of *xre-res*$^{Pp}$ to repress transcription, suggesting that the imperfectness of the inverted repeat is required for optimal transcription but not repression (Fig 2D).

## The Xre-RES$^{Pp}$ complex binds directly to S4 *in vitro*

The presence of an HTH domain in Xre and the observed repression of transcription upon expression of *xre-res*$^{Pp}$ indicated that the Xre-RES$^{Pp}$ protein complex might bind directly to DNA. To investigate this, we analysed the binding between the Xre-RES$^{Pp}$ protein complex and different parts of the P$_{XR}$ promoter *in vitro*. First, we expressed and purified the intact Xre-RES$^{Pp}$ complex via a N-terminal His$_6$-tag on the RES toxin (Xre-RES$^{Pp}$$_{His6}$) and used size exclusion chromatography coupled to multi-angle light scattering (SEC-MALS) to show that Xre and RES$_{His6}$ form a protein complex with an estimated

MW of 92 ± 4 kDa (Fig 3A), which best matches the theoretical mass of the 4:2 Xre-RES complex of approximately 103 kDa observed in the crystal structure (Fig 3B) [14]. We then analysed the ability of the purified complex to bind a set of 30 bp dsDNA probes each covering one of the four sequence elements (S1 to S4) in the $P_{XR}$ promoter (Figs 1A and S1A). For this, purified Xre-RES$^{Pp}_{His6}$ complex and dsDNA probes were incubated in 1:1 molar ratio followed by analytical SEC (a-SEC). A significant change in elution volume of 0.2 mL and a concomitant change in the 260:280 nm absorbance ratio from 0.54 to 0.68 was observed for complex mixed with S4 dsDNA (Fig 3C), but not for S1, S2 or S3 dsDNA (S3A-C Fig), suggesting that the Xre-RES$^{Pp}$ complex binds specifically to the S4 element of the $P_{XR}$ promoter.

Previously, structure determination of the Xre-RES$^{Pp}$ complex has shown that the 4:2 heterohexamer contains two complete Cro-like Xre$^{Pp}$ dimers [14], each potentially capable of binding dsDNA via interaction with adjacent major grooves [27]. To investigate whether the S4 element is the only binding site in the promoter, we determined the binding stoichiometry by analysing the binding at different protein:DNA ratios using SEC-MALS. At a 1:1 ratio of Xre-RES$^{Pp}$ complex to dsDNA duplex, and thus a ratio of putative binding sites to DNA of 2:1, we found that the protein complex was unable

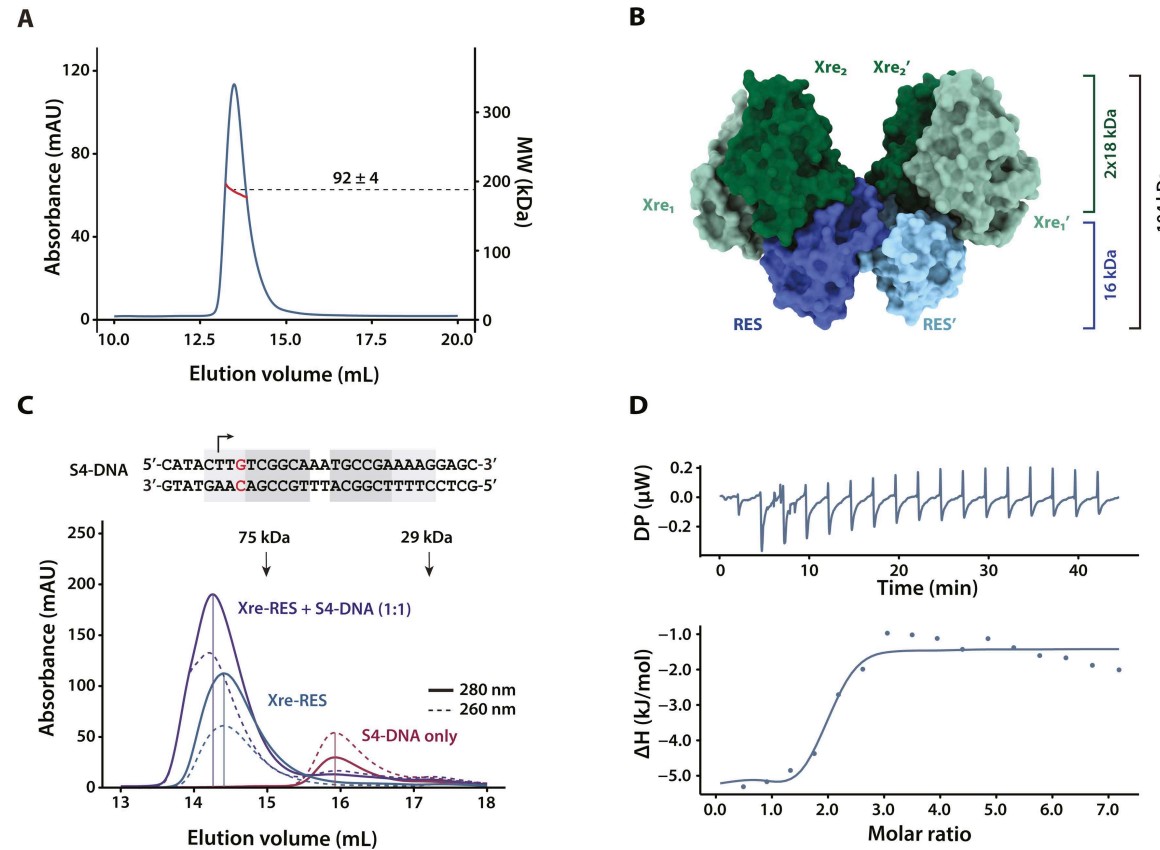

**Fig 3. Xre-RES$^{Pp}$ forms a 4:2 hexameric complex in solution and interacts specifically with the S4 element of the promoter.** (A) SEC-MALS analysis of purified Xre-RES$^{Pp}_{His6}$ complex. The chromatogram shows the elution profile as measured by absorption at 280 nm (blue) and the molecular mass measured by MALS (red). The average mass corresponding to the middle of the peak is 92 ± 4 kDa (dotted line). (B) The crystal structure of heterohexameric Xre-RES$^{Pp}$ (PDB: 6GW6) with total mass (103 kDa) and HTH domains indicated [14]. The RES toxins are coloured in shades of blue, while the Xre antitoxins are in shades of green. (C) Analytical SEC (a-SEC) analysis of Xre-RES$^{Pp}_{His6}$ (Xre-RES, blue), S4 (S4-DNA only, red), and DNA mixed with protein in a 1:1 ratio (Xre-RES + S4-DNA, purple). Black arrows represent known molecular weight standards (conalbumin, 75 kDa; carbonic anhydrase, 29 kDa). The full line shows 280 nm absorbance, while the dotted line is 260 nm absorbance. (D) Isothermal Titration Calorimetry (ITC) titration of S4 dsDNA into the Xre-RES$^{Pp}$ complex. The top panel shows the raw thermogram (DP, differential power), while the bottom panel shows the integrated heat per injection (ΔH, kJ/mol). The solid is line is the fit using the "one set of sites" model.

to bind all DNA present in solution (S3D Fig), and increasing this to a 1:2 ratio of protein complex to DNA, no additional shift in the elution volume or calculated mass was observed as compared to the 1:1 ratio (S3D Fig). To further probe the stoichiometry of binding between the Xre-RES$^{Pp}$ complex and S4 dsDNA, we saturated the protein complex with increasing amounts of DNA and used high-resolution SEC (2.4 mL) to check for indication of a second DNA binding site. Even at a molar ratio of protein complex to dsDNA of 1:10, only a marginal shift in the elution volume of 0.01 ml (0.6% CV) was observed (S3E Fig). Together, this suggests that the 4:2 Xre-RES$^{Pp}$ complex only binds a single copy of S4 dsDNA. To further quantify the binding interaction between the Xre-RES$^{Pp}$ complex and S4, we determined the binding coefficient ($K_D$) using isothermal titration calorimetry (ITC). This showed that the Xre-RES$^{Pp}$ complex binds with a 1:1 ratio and high affinity to the native S4 element ($K_D = 231 \pm 46$ nM, Fig 3D), while the affinities towards both the 5'-perf ($K_D = 341 \pm 264$ µM) and 3'-perf variants ($K_D = 80 \pm 79$ µM) of S4 were drastically reduced (Table 1). Given that repression by *xre-res*$^{Pp}$ is intact for these two variants (5'-perf and 3'-perf) *in vivo*, these data are consistent with the existence of a second DNA binding site that strengthens the protein-DNA interaction in the context of the full-length promoter.

### The structure of the Xre-RES$^{Pp}$-DNA complex reveals an unusual, asymmetrical DNA binding site

To get a molecular understanding of the interaction between Xre-RESPp and promoter DNA, we determined the crystal structure of the Xre-RESPp complex bound to a 30 bp dsDNA oligo covering the S4 element to 2.7 Å (R/ R$_{free}$ = 22.8%/ 26.9%, Fig 4A and Table 2). The structure contains four complete heterohexameric (4:2) Xre-RES$^{Pp}$ complexes and four DNA duplexes in the crystallographic asymmetric unit (S4A Fig). In all molecules, one Xre dimer of each Xre-RES$^{Pp}$ complex interacts strongly with two adjacent major grooves on dsDNA (Figs 4A and S4B). The interaction between the Xre dimer and DNA causes the major groove on the opposing side of the double helix to expand from 11.8 Å in canonical B-DNA to 14 Å (Fig 4A). Significantly, we observe that the Xre dimer is asymmetrical in the way it interacts with DNA and overall approaches the double helix in a different way than classical HTH domains such as the Cro-repressor [28]. More specifically, the angle between the helical axis of DNA and that of the recognition helix of the outer Xre molecule is 61.4° while it is 57.5° for the inner Xre molecule. These values are both significantly smaller and more different from each other than observed for the canonical structure of the Cro repressor bound to DNA (PDB 1RPE), in which the two recognition helices of the Cro dimer approach DNA at nearly identical angles of 68.8° (S4C Fig) (Shimon & Harrison, 1993). Several residues in Xre, in particular Arg60, Arg67, Thr68, Lys70, and Arg72, interact tightly with C13, C20, and G21 on the (+) strand, as well as T7, T8, T9, and A14 on the (-) strand of S4 DNA (Fig 4B and 4C). The dominant interactions are centred around the conserved Arg67 of the innermost Xre, which engages in hydrogen bonds to the N3 atom of T9 of the DNA (-) strand (Fig 4C). Interestingly, Arg67 of the outermost Xre displaces A14 of the (+) strand and forms hydrogen bonds to the N3 of C13 (+) (Fig 4C). Arg60, Thr68, Lys70, and Arg72 also make non-sequence specific interactions to the phosphodiester backbone of DNA. Of the interacting residues, both Thr68 and Arg72 are functionally conserved across *xre-res* loci from a wide range of bacterial species, while Arg67 and Lys70 appear conserved in a subset of species (Fig 4D). To support the importance of Arg67 in DNA binding by Xre, we substituted this residue with alanine (R67A) and found that transcriptional repression by *xre*$^{R67A}$-*res in vivo* was abolished (Figs 4E and S4D).

**Table 1. Isothermal Titration Calorimetry (ITC).**

| dsDNA | K$_D$ [M] | SEM | Enthalpy (H) [kJ/mol] | Entropy (-ST) [J/mol] | Free energy (G) [kJ/mol] |
|---|---|---|---|---|---|
| S4 | 2.31·10$^{-7}$ | 4.61·10$^{-8}$ | -3.35 ± 1.29 | -31.8 ± 1.66 | -38.4 ± 0.71 |
| 5'-perf | 3.41·10$^{-4}$ | 2.64·10$^{-4}$ | -5.36 ± 0.92 | -26 ± 2.09 | -31.3 ± 1.25 |
| 3'-perf | 8.06·10$^{-5}$ | 7.97·10$^{-5}$ | 9.14 ± 7.28 | -42.9 ± 8.06 | -33.8 ± 0.91 |

S4 dsDNA, or variants, was titrated into Xre-RES$^{Pp}$ complex. The calculated binding affinities (K$_D$), the enthalpy (H), the entropy with temperature (-ST), and the Gibbs free energy (G) are shown. All values are means of three (n = 3) independent replicates with standard error of mean (SEM) indicated.

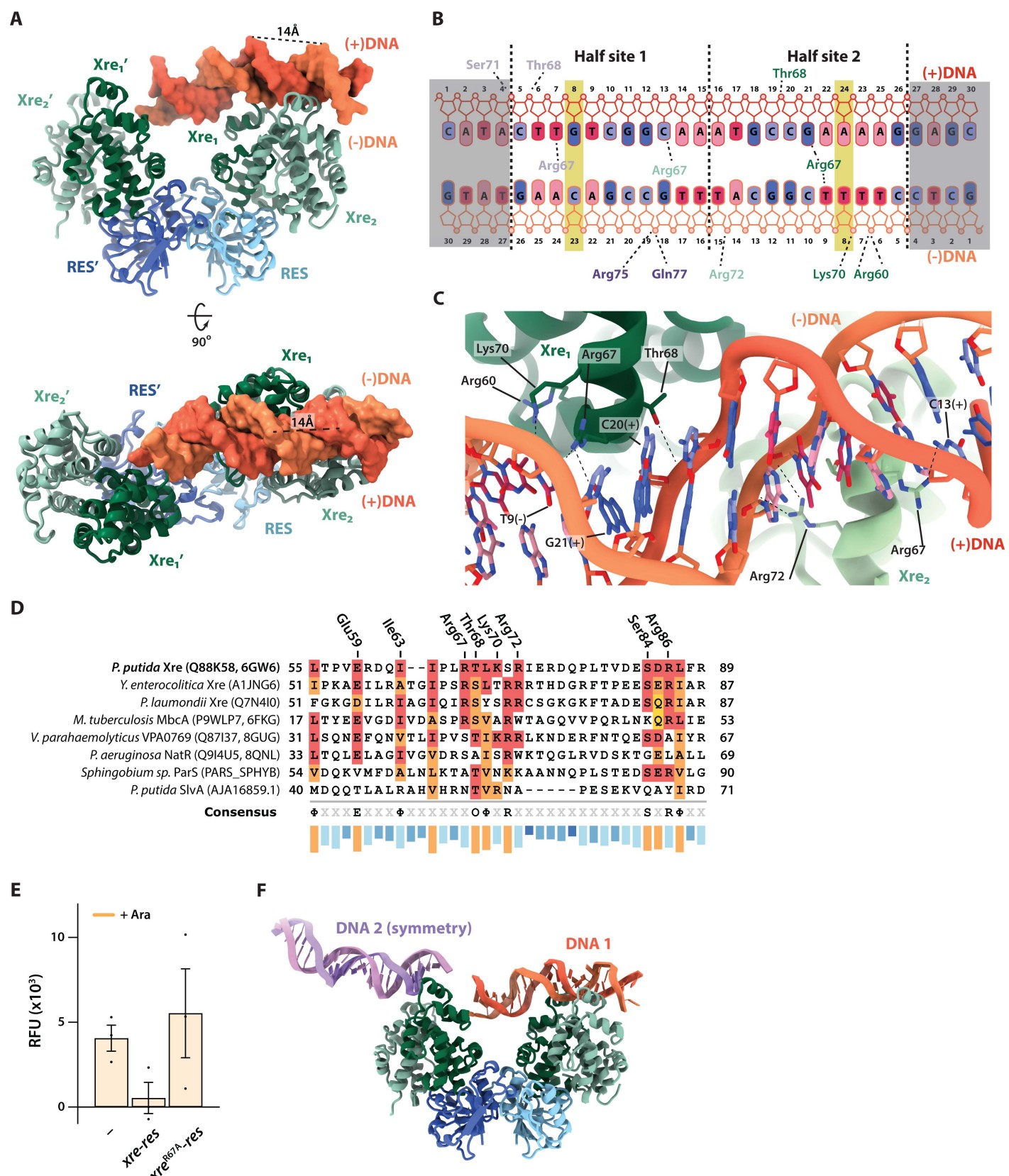

**Fig 4. Crystal structure of the DNA-bound Xre-RESPp complex.** (A) Overview of the *P. putida* Xre-RES complex (green/blue cartoon) is bound to a 30 bp dsDNA (red/orange) with both HTH recognition helices of the of a single Xre dimer interacting with DNA. (B) Schematic overview of the interactions between Xre and DNA including both sequence specific and unspecific contacts as well as the contacts from the symmetry related interaction. The shaded DNA region could not be build. (C) Close-up on the interaction between Xre and DNA with interactions between selected residues of both Xre molecules and the DNA highlighted. (D) Partial sequence alignment of *P. putida* Xre and a range of homologous proteins from other bacterial species. UniProt IDs are listed in parentheses for all proteins, along with PDB IDs for *P. putida* Xre (PDB ID 6GW6) [14], *M. tuberculosis* MbcA (PDB ID 6FKG) [12], *V. parahaemolyticus* VPA0770 (PDB ID 8GUG) [18], and *P. aeruginosa* NatR (PDB ID 8QNL) [19]. Residues conserved with *P. putida* Xre are shown in red with functionally similar residues in orange. Starting and ending residue numbers are listed at each end of the sequences. (E) Promoter activity in *E. coli* MG1655 using promoter reporter plasmid pGH254Kgfp::$P_{XR}$ ($P_{XR}$) and either pBAD33 (-), pBAD33::*xre-res*$^{Pp}$ (*xre-res*), or pBAD33::*xre*$^{R67A}$-*res*$^{Pp}$ (*xre*$^{R67A}$-*res*), as indicated. Complete growth curves and details of the constructs are found in S4 Fig. (F) In the crystal, the Xre-RES complex (blue/green) interacts with a primary dsDNA molecule (DNA 1) as well as a symmetry-related DNA molecule (DNA 2).

**Table 2. Data collection and refinement statistics (molecular replacement).**

| | DNA-bound Xre-RES$^{Pp}$ complex (PDB: 9R35) |
|---|---|
| **Data collection** | |
| Space group | P 1 2₁ 1 |
| Cell dimensions | |
| *a, b, c* (Å) | 81.23, 185.11, 170.79 |
| α,β,γ (°) | 90, 90.01, 90 |
| Resolution (Å) | 26 – 2.7 (2.77 – 2.70)* |
| $R_{sym}$ (%) | 9.2 (122.7) |
| *I* / σ*I* | 8.29 (0.85) |
| Completeness (%) | 90.3 (83.4) |
| Redundancy | 3.4 (3.0) |
| CC½ (%) | 99.7 (35.8) |
| **Refinement** | |
| Resolution (Å) | 26 – 2.7 (2.77 – 2.70) |
| No. of reflections | 146,206 (7846) |
| $R_{work}$ / $R_{free}$ (%) | 22.8 (35.2)/ 26.9 (36.5) |
| No. of atoms | 31,814 |
| Protein | 31,524 |
| Solvent | 65 |
| Average *B*-factors (Å²) | |
| Protein | 89.1 |
| Solvent | 79.8 |
| RMS deviations from ideality | |
| Bond lengths (Å) | 0.004 |
| Bond angles (°) | 0.628 |
| Rotamers (%) | |
| Favoured | 99.6 |
| Outlier | 0.4 |
| Ramachandran values (%) | |
| Favoured | 98.1 |
| Allowed | 1.5 |
| Outlier | 0.4 |

*Values in parentheses represent highest-resolution shell.

Intriguingly, inspection of the second Xre dimer of the Xre-RES heterohexamer reveals that there are interactions to one end of a DNA duplex from a symmetry-related complex (Fig 4F). The specific interactions made at this site vary slightly between complexes in the asymmetric unit but in all cases include base-specific interactions between Arg67 of the outermost Xre molecule to O2 on T7 on the (+) strand, and non-sequence specific interactions between Thr68 and Ser71 to the backbone of A4 and C5 of the (+) strand as well as between Arg75 and Gln77 and C19 of the (-) strand (Fig 4B). In summary, the structure of DNA-bound Xre-RES$^{Pp}$ confirms that one of the two Xre dimers in the complex recognises S4 dsDNA using base-specific interactions dependent on Arg67 and moreover, that the second Xre dimer is capable of simultaneously interacting with DNA supporting the idea that there might be a secondary DNA binding site inside the full operator region.

**The Xre$^{Pp}$ dimer is stabilised by higher-order complex formation and required for transcriptional repression**

Since the structure shows that only one of the two Xre dimers in the complex is bound to S4 dsDNA, we speculated whether Xre alone can repress transcription. To investigate this, we used the transcriptional coupling to GFP to compare the activity of the wild type P$_{XR}$ promoter during induction of either *xre$^{Pp}$* or *xre-res$^{Pp}$*. While expression of the complex fully repressed transcription as seen before, expression of *xre$^{Pp}$* alone only partially, and not significantly, repressed transcription (Fig 5A). Moreover, we note that the promoter activity between biological replicates was considerably more variable upon expression of *xre$^{Pp}$* (Figs S5A and 5B), indicating that the Xre$^{Pp}$:DNA interaction is unstable in the absence of RES$^{Pp}$. To assess if the weak repression upon induction of *xre$^{Pp}$* alone is due to lack of formation of a stable Xre dimer, we expressed and purified the isolated Xre$^{Pp}$ antitoxin with a C-terminal His-tag (Xre$^{Pp}$$_{CHis6}$) to avoid disturbing the dimerisation interface. Purified Xre$^{Pp}$$_{CHis6}$ was very prone to aggregation and could not be concentrated above 10 µM, supporting that complex formation contributes to stabilising the antitoxin. Moreover, the mass of purified Xre$^{Pp}$$_{CHis6}$ antitoxin was determined by SEC-MALS to 22±1 kDa, corresponding to a monomer of Xre$^{Pp}$ (Fig 5B). In summary, we conclude that the presence of RES$^{Pp}$ is required for formation of a functional and DNA-binding competent Xre$^{Pp}$ dimer.

To further investigate the stability the Xre$^{Pp}$ dimer within the complex, mass photometry (MP), which is a single-particle mass-determination method, was used to determine the mass of the purified Xre-RES$^{Pp}$ complex. The particle distribution was not uniform and at 10 nM we observed two populations of which the larger and most abundant was around ~90 kDa and the smaller was around 76 kDa. At 5 nM, the same two populations were observed, but with the smaller MW complex being most abundant, however an additional peak around 50 kDa was also seen. Measuring at 2 nM revealed a mostly homogeneous peak at with 50 kDa, with a trailing tail of larger particles. This concentration dependent change in mass distribution and median masses (87±15 kDa, 76±14 kDa, 54±14 kDa) corresponds roughly to a transition from the 4:2 (103 kDa) complex via a 3:2 (86 kDa) to a 2:2 (69 kDa) complex (Fig 5C). Together, this suggests that even in presence of RES, the Xre$^{Pp}$ dimer is unstable and the Xre-RES$^{Pp}$ complex can exist on both 4:2, 3:2, and 2:2 forms. To confirm the existence of the 2:2 complex *in vitro*, we expressed and purified a variant of the Xre-RES$^{Pp}$ complex with a N-terminal His$_6$-tag on Xre (Xre$_{NHis6}$-RES$^{Pp}$), which, based on the crystal structure, should prevent Xre dimerisation. The expressed protein was highly prone to precipitation during purification, consistent with a destabilised Xre$^{Pp}$ dimer that would result in free and unstable Xre$^{Pp}$ monomers as well as a 2:2 complex. This was supported by SEC, which showed two peaks corresponding to the 2:2 complex and isolated Xre$^{Pp}$ antitoxin, respectively (S6A and 6B Figs). The successful isolation of a 2:2 complex was further confirmed by mass photometry, where the main protein population at both 10 nM and 2 nM exhibited masses of 63±18 kDa (Fig 5D).

As the isolated Xre$^{Pp}$ antitoxin is a monomer and induction of *xre$^{Pp}$* alone only weakly represses transcription *in vivo*, we predicted that the isolated 2:2 complex, containing two Xre monomers, would also have reduced DNA affinity. Analysis by high-resolution SEC column of samples containing only S4 DNA, only the 2:2 complex, or 2:2 complex and S4 DNA in ratio of 1:10 showed that the elution volume of the 2:2 complex is not affected by the addition of DNA (Fig 5E). This supports that the 2:2 is a non-binding form of the Xre-RES$^{Pp}$ complex and thus a model in which the complex exists in a concentration-dependent equilibrium between non-binding (2:2) and DNA-binding (4:2) forms.

PLOS Genetics

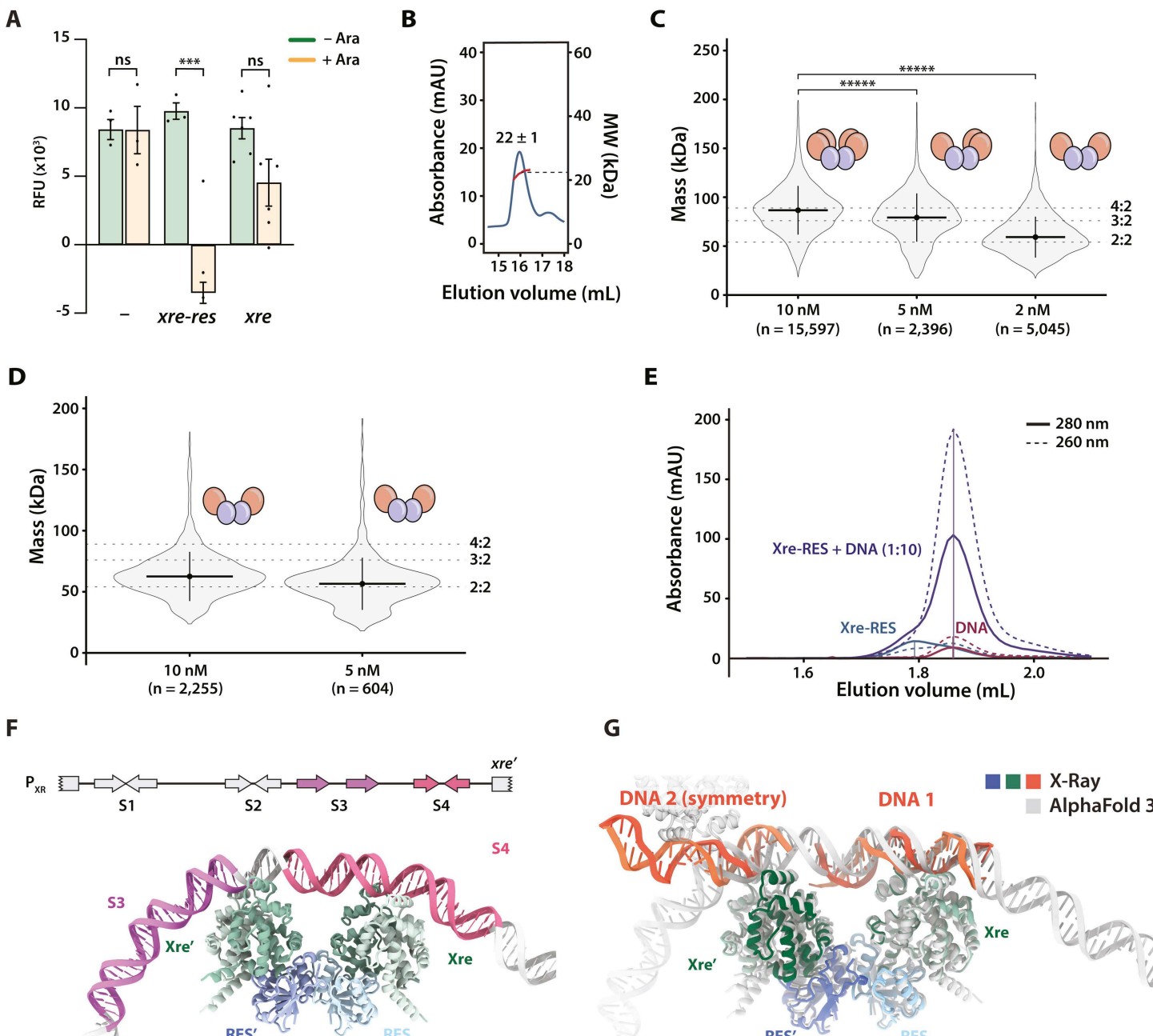

**Fig 5. DNA binding requires the Xre dimer, which is unstable at low concentrations.** (A) Promoter activity assays in *E. coli* MG1655, where the promoter reporter plasmid pGH254Kgfp::$P_{XR}$ ($P_{XR}$) was combined with either empty pBAD33 (-), pBAD33::*xre*$^{Pp}$ (*xre*), or pBAD33::*xre-res*$^{Pp}$ vector (*xre-res*), as indicated. Complete growth curves and details of the constructs are found in S4 Fig. Statistical analysis was performed using Welch's t-test (with equal variances). *ns*, not significant; ***, $p < 0.001$. (B) SEC-MALS analysis showing that isolated Xre$^{Pp}_{CHis6}$ is a monomer with an approximate mass of $22 \pm 1$ kDa in solution. The chromatogram shows elution profile measured by 280 nm absorbance (blue) and calculated mass from MALS (red). (C) Single-molecule mass determination of Xre$^{Pp}$-RES$^{Pp}_{His6}$ at 2, 5, and 10 nM. The violin plots represent the density of all measured single-molecule events. The solid black horizontal line indicates the median mass, while the vertical line shows the standard deviation. The theoretical masses of the Xre$^{Pp}$-RES$^{Pp}_{His6}$ complex with 4:2, 3:2, and 2:2 stoichiometries are indicated by dotted lines. Significance was tested with the Wilcoxon signed rank test. *****, $p < 5.55e\text{-}7$. (D) Single-molecule mass determination for Xre$_{His6}$-RES$^{Pp}$ on the 2:2 complex form at 5 and 10 nM, as in (C). (E) High resolution analytical SEC (a-SEC) analysis of Xre$_{NHis6}$-RES (Xre-RES, blue), S4 (DNA, red), and DNA mixed with protein in a 1:10 ratio (Xre-RES + DNA, purple). The full line shows 280 nm absorbance, and the dotted line is 260 nm absorbance. (F) AlphaFold3-predicted structure of the Xre-RES complex interacting with the full 172 bp promoter region (not all nucleotides shown). (G) Alignment of crystal structure of the DNA-bound Xre-RES complex, including the symmetry-related dsDNA molecule (blue/green/orange) with the AlphaFold 3 prediction (grey).

## Structural basis for Xre-RES<sup>Pp</sup> binding at two sites in the P<sub>XR</sub> promoter region

Since the crystal structure suggests that both DNA-binding Xre dimers on a Xre-RES heterohexamer can engage in DNA binding simultaneously, we decided to use AlphaFold 3 (AF3) to predict the interaction between Xre-RES and the full 172 bp promoter [29]. The resulting prediction showed high confidence for the Xre-RES protein complex (average pLDDT 90.1), with a more varying confidence for the DNA having pLLDTs between 24.2 and 81.5 (average 53.9). The entire prediction has a PTM score of 0.64 which, while low, is unsurprising given the variance in pLDDT and the length of free DNA. The interaction has an iPTM score of 0.6, which is in the grey zone for confidence and thus would require additional support for validation [29]. This support is provided by comparison to the crystal structure, as the predicted model has both Xre dimers engaged in DNA binding about 20-bp apart (S7A-C Fig), with one site closely matching the primary S4 DNA binding site observed in the crystal structure and the other located upstream overlapping partly with S3 as well as involving the region between S3 and S4 (Fig 5F). Comparison of the prediction to the crystal structure reveals a good correspondence in the details of the interaction at the primary site, including the central of Arg67 (S7D Fig). We consider it, in itself, remarkable that AF3 is able to precisely map the binding site of a bacterial transcription factor within such a long stretch of DNA with nucleotide precision, especially given that there are no structures of similar protein complexes bound to DNA available. However, while the prediction gets the location of the interaction site right and correctly predicts the unusual, asymmetrical way the Xre recognition helices bind the DNA, it does not capture all the subtle details of the deformation of the DNA that take place upon induced fit (Fig 5F). Remarkably, however, the predicted interaction of Xre-RESPp with DNA at the secondary binding site located upstream at the S3-S4 interface is strikingly similar to the interaction to a symmetry-related S4 DNA molecules in the crystal structure. We consider this as strong support for the validity of the overall prediction given the complementarity of the two techniques (Fig 5G). The agreement between the crystal structure and the *in silico* prediction suggests that the interaction between the second Xre dimer and the symmetry-related DNA molecules reflect a true biological ability of the complex to interact with higher-order DNA at two sites simultaneously. Collectively, this analysis allows us to put forward a detailed and testable model for how the Xre-RESPp complex interacts with higher-order promoter DNA via the S3 and S4 repetitive elements.

## Discussion

In this paper, we present a detailed model for transcriptional autoregulation of the *P. putida xre-res* operon. We show that the heterohexameric Xre-RES<sup>Pp</sup> protein complex efficiently represses transcription from the *xre-res* locus through direct binding to an imperfect inverted repeat region (S4) in the promoter region, and the structure of the Xre-RES<sup>Pp</sup> complex bound to this DNA motif provides a detailed molecular view of how the promoter is specifically recognised. We also confirm that the unusual 4:2 stoichiometry of the TA complex observed in the isolated crystal structure [14], in which each of the toxins of a central dimer interacts with a dimer of antitoxins, is present in solution, and further show that isolated Xre<sup>Pp</sup> antitoxin is a monomer that only weakly represses transcription *in vivo*. This suggests that formation of the Xre dimer, which is stabilised through interactions with the RES toxin, is necessary for efficient binding to DNA. Supportive of this, a 2:2 complex present at low concentrations does not bind DNA. Together, these data allow us to propose a model for transcriptional regulation of the *xre-res*<sup>Pp</sup> operon (Fig 6). The expressed 4:2 Xre-RES<sup>Pp</sup> complex represses transcription through binding via one Xre dimer to the S4 imperfect repeat, followed by binding of the second Xre dimer to the region between S3 and S4. This serves to control transcription and maintain a low cellular concentration of RES toxin. But since the RES-bound Xre-dimer is unstable, dissociation and formation a 2:2 complex will occur over time. In this 2:2 complex, Xre still binds to, and neutralises, the RES toxin, but it is unable to bind DNA and consequently, transcription of the *xre-res*<sup>Pp</sup> operon is eventually reactivated. This increases the cellular concentrations of the Xre antitoxin, leading to the formation of a repressing DNA-binding 4:2 complex, possibly via the 3:2 intermediate.

This model resembles the recently proposed translation-responsive model [30], which states that the expression of TA systems is primarily regulated by the rate of antitoxin production. This model assumes that degradation of antitoxin over

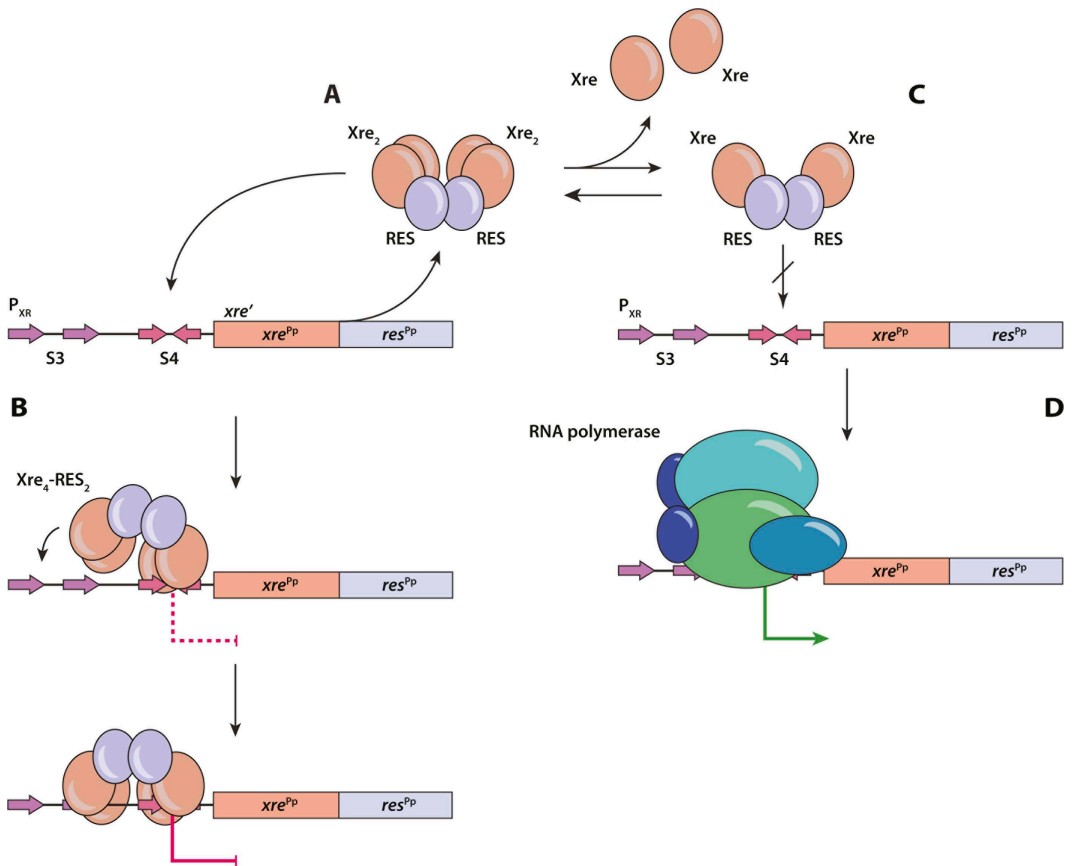

**Fig 6. Model for transcriptional regulation of the *xre-res*^Pp operon.** Expression of *xre-res*^Pp leads to the formation of a 4:2 Xre-RES^Pp complex (A), in which one Xre dimer binds an imperfect inverted repeat in the promoter region (S4), followed binding of the second Xre dimer to the region between S3 and S4 and subsequently transcriptional repression (B). This lowers the concentration of the Xre antitoxin, leading to the dissociation of the Xre dimer (C) and formation of a toxin-neutralising, but non-DNA binding 2:2 complex (D), and thereby initiation of transcription by the RNA polymerase.

time is constant and that the net level of antitoxin thus is controlled by transcriptional and translational activity of the TA operon. Here, we propose that transcriptional autoregulation is based on the concentration of the Xre antitoxin, which again depends on Xre expression from the *xre-res*^Pp operon. Interestingly, the structure of the Xre-RES complex from *V. parahaemolyticus* (VPA0770-VPA0769) also exists in a heterohexameric (4:2) form that exhibits a W-shaped architecture like Xre-RES^Pp and moreover, the antitoxin was also found in an equilibrium between a dimer and a monomer [18]. The VPA0770-VPA0769 complex was found to form a hetero-dodecameric complex (4:8) at high protein concentrations (≥ 5 mg/ml), which was proposed to enhance the stability of the TA complex and thereby facilitate tighter regulation of toxin activity. However, at biologically relevant levels (≤ 1 mg/ml), the complex dissociates into heterohexamers similar to Xre-RES^Pp. Additionally, higher-order structural analysis using PISA [31] indicates that the most stable assemblies in solution are either a 2:2 or a 2:4 form. These observations suggest that the *VPA0770-VPA0769* system might operate in a manner similar with the model proposed here. The existence of a dynamic equilibrium between two oligomeric forms of the Xre-RES complex has also been described for the Xre-RES complex NatRT from *P. aeruginosa* [19], which forms a 4:2 inert state and a 2:1 toxic state and in line with the present findings, the 4:2 NatRT complex autoregulates transcription of *natR-natT*, while the 2:1 state is unable to bind the promoter region and thus repress transcription. However, in this case, the interaction surface between NatR and NatT is mediated by a helix-loop-helix domain (referred to as the 'Flap')

not found in RES$^{Pp}$, and moreover, the switch between the two oligomeric states of NarRT is due to disintegration of the NatT toxin dimer, which we do not observe. Finally, the activity of the NatT toxin affects how the complex binds the promoter, and thus how it regulates transcription. We have not been able to detect NADase activity of RES$^{Pp}$ when bound to Xre, suggesting that the RES domain toxins are regulated and activated by different mechanisms. Taken together, while the details of the mechanisms of Xre-RES systems differ between organisms, the existence of a dynamic equilibrium between oligomeric states, each playing a unique role in activity or regulation of gene expression, appears to apply more generally across Xre-RES homologs.

The crystal structure of the Xre-RES$^{Pp}$ complex bound to promoter DNA reveals an unusual structural asymmetry in the way the Xre HTH recognition helix interacts with DNA when compared to canonical HTH-containing protein repressors, such as the bacteriophage lambda Cro repressor [20]. This, together with the preference for the imperfect inverted repeat in S4, suggests that asymmetry is an important feature for optimal protein-DNA interaction. We note that the bending of the DNA as well as asymmetrical interaction between Xre and the S4 element optimises the involvement of Arg67. The importance of this is supported by the reduced binding affinity observed for Xre-RES$^{Pp}$ towards the 5'-perf and 3'-perf S4 variants *in vitro* (Table 1). In addition, the asymmetry of the two recognition helices involved in DNA binding ensures specificity as well as directional binding of the protein complex, positioning the second 'free' Xre dimer to bind a potential secondary operator site. While *in vitro* functional data show that a single Xre-RES$^{Pp}$ complex binds only one copy of S4 dsDNA, deletion of the central palindromic motif of S4 (Δrepeat) within the full $P_{XR}$ promoter, including bases identified by our structural data as critical for specific interaction with Arg67, still results in partial repression by *xre-res*$^{Pp}$ *in vivo*. This finding suggests the existence of a second binding site within the full promoter and supportive of this, the AF3 prediction indicates that the second Xre dimer binds between S3 and S4 in the promoter, in a manner similar to the observed secondary interaction in the crystal. We believe that this interaction is highly significant because crystal contacts often reveal important functional aspects of the crystallised macromolecules. However, these can be hard to validate and separate from mere packing contacts and likewise, hypothetical interactions in predicted models need validation before any conclusions can be drawn. Here, the results of two completely independent techniques (AF3 and x-ray crystallography) strongly support each other in suggesting both the location and specific mode of interaction at the secondary site.

In conclusion, transcriptional regulation of the *xre-res*$^{Pp}$ operon relies on an asymmetrical interaction of the symmetrical Xre-RES$^{Pp}$ protein complex to the operator site, as well as on a dynamic, concentration-dependent stoichiometry of the Xre-RES$^{Pp}$ complex. Together, this puts forward a detailed and testable model for how bacterial transcription factors interacts with higher-order promoter DNA (Fig 6). This model demonstrates how a bacterial protein repressor and its operator sites co-evolve to ensure optimal and specific transcriptional regulation and the presence of two DNA-binding domains allows for a tighter transcriptional repression through avidity, making the system more robust towards both perturbations and mutations.

## Materials and methods

### Bacterial strains and plasmids

Bacterial strains and plasmids used in this study are listed in S2 Table, and DNA oligonucleotides in S3 Table. Detailed descriptions of plasmid construction are found in S1 Text. For site-directed mutagenesis, the Q5 Site-Directed Mutagenesis Kit from NEB was used according to the protocol and mutagenesis primers were designed using NEBaseChanger.

### Measurements of promoter activities and *in vivo* repression

For promoter activity assays, *E. coli* MG1655 was co-transformed with transcriptional fusion vector pGH254Kgfp and a pBAD33 vector. Overnight cultures were grown in M9 medium supplemented with 0.2% Casamino acids, 0.4% glycerol, 50 mg/ml chloramphenicol, 25 mg/ml kanamycin and 0.2% glucose. Overnight cultures were washed once in M9 medium

and diluted to 0.03 in M9 medium supplemented with 0.2% Casamino acids, 0.4% glycerol, 50 µg/ml chloramphenicol and 25µg/ml kanamycin. From this, cultures were split into non-induced or induced (by 0.2% arabinose) sub-cultures. Samples of 200 µl were grown in triplicates in a black/clear bottom 96-well plates (Thermo Scientific, cat. no. 165305), using a Synergy H1 microplate reader (BioTek). Optical density (600 nm) and fluorescence intensity (excitation, 479 nm; emission, 520 nm) was measured every 5 min for 12 h. A strain with empty pGH254Kgfp and pBAD33 was used as control and fluorescence intensity at a specific time point was calculated as described by [32], using the formula $I(t) = I_u(t) - \frac{A(t)}{B(t)} I_b(t)$ where $I(t)$ is the fluorescence intensity at a specific time point, $I_u(t)$ is the uncorrected intensity, $A(t)/B(t)$ is the absorbance ration between the measured culture and the culture used as blank, and $I_b(t)$ is the fluorescence intensity of the culture used as blank. The calculated fluorescent intensity was divided by $OD_{600}$ to obtain corrected fluorescence intensity per $OD_{600}$ unit relative to the control strain (relative fluorescence units, RFU). The assay was repeated three times for each strain. Mean values were calculated for technical triplicates and used to calculate a mean-of-mean values and standard mean of error (SEM) of biological triplicates. Statistical analysis was performed in R using Welch's t-test with the assumption of equal variance. Graphs were prepared with Prism.

## Protein expression and purification

For protein expression, plasmids pETDuet::$res_{NHis6}$-$xre^{Pp}$, pETDuet::$xre_{NHis6}$-$res^{Pp}$, or pET-29b(+)::$xre_{CHis6}{}^{Pp}$ were transformed into *E. coli* BL21(DE3). Cells were grown to an $OD_{600}$ of 0.6-0.8, induced by the addition of 0.5 mM IPTG for 20–22 h at 20°C, and harvested by centrifugation (6,000 rpm, 10 min, 4°C).

For purification of the Xre-RES$^{Pp}{}_{NHis6}$ and the Xre$_{NHis6}$-RES$^{Pp}$ complex, cell pellets were resuspended in lysis buffer (50 mM Tris-HCl, pH 8.5, 300 mM KCl, 10 mM imidazole, 3 mM β-mercaptoethanol (BME)) supplemented with 1 ng/µl DNase I and 1 mM phenylmethylsulfonyl fluoride (PMSF). Cells were lysed by high-pressure homogenization (Avestin) at 15,000 psi, and clarified by centrifugation (14,000 rpm, 45 min, 4°C). Clarified cell lysates were loaded onto a 1 ml HisTrap FF column (Cytiva) and columns were washed with lysis buffer including 40 mM imidazole. Protein complexes were eluted with elution buffer (50 mM Tris-HCl, pH 8.5, 150 mM KCl, 300 mM imidazole) and further purified using a 1 ml Source 15Q column (GE Healthcare), running in buffer A (50 mM Tris-HCl, pH 8.5, 5 mM (BME) and a gradient into buffer B (50 mM Tris-HCl, pH 8.5, 5 mM BME, 1 M KCl). Final separation was achieved on a Superdex 200 Increase 10/300 GL column (GE Healthcare) equilibrated in gel filtration buffer (20 mM HEPES-KOH, pH 7.5, 150 mM KCl, 5 mM MgCl$_2$, 3 mM BME).

For purification of Xre alone (Xre$_{CHis6}{}^{Pp}$), cell pellets were resuspended in lysis buffer (50 mM HEPES-KOH, pH 7.5, 400 mM KCl, 10 mM imidazole) supplemented with 1 ng/µl DNase I and 1 mM PMSF, lysed by high-pressure homogenization (Avestin) at 15,000 psi, and clarified by centrifugation (14,000 rpm, 45 min, 4°C). Clarified cell lysate was loaded onto a 1 ml HisTrap FF column (Cytiva), washed with lysis buffer including 30 mM imidazole and eluted using 300 mM imidazole. Eluate was precipitated overnight at room temperature with 45% ammonium sulphate (AmS) and clarified by centrifugation (20.000 rpm, 30 min, 20° C). Supernatant was collected and further precipitated with 60% AmS overnight. Precipitated protein was collected by centrifugation (20.000 rpm, 30 min, 20° C) and resuspended in gel filtration buffer (20 mM HEPES-KOH, pH 7.5, 150 mM KCl, 3 mM BME) until conductivity was 15 mS/cm. Final separation was achieved on a Superdex 75 10/300 GL column (GE Healthcare) equilibrated in gel filtration buffer.

## Analytical size exclusion chromatography

To screen the interaction between the Xre-RESPp complex and putative binding sites within the promoter, 30 bp dsDNA oligos corresponding to S1, S2, S3 or S4 in the promoter region (Figs 1A and S1A) were made using oligos specified in S3 Table. Briefly, equimolar amounts of complementary DNA oligos were mixed in milliQ water heated to 95°C and cooling to room temperature. Concentrations were determined by nanodrop, using calculated extinction coefficients, and 1.5 nmoles of dsDNA and/or purified Xre-RES$^{Pp}{}_{NHis6}$ each were incubated for 5 min at 37° C. Samples containing protein alone,

DNA alone, or mixed protein and DNA were analysed by analytical gel filtration (a-SEC) using a Superdex 200 Increase 10/300 GL column (GE Healthcare) running in DNA binding buffer (20 mM HEPES pH 7.5, 150 mM KCl, 5 mM MgCl$_2$ and 3 mM BME). This method probes only high strength (low $K_D$) interactions which would be necessary for transcriptional control. To assess stoichiometry of protein to DNA in the interaction, samples with protein to DNA ratios of 2:1, 1:2, and 1:10 were analyzed by a-SEC and SEC-Multi angle light scattering (MALS) in DNA binding buffer. For 1:2 and 1:10 ratios of protein to DNA, a Superdex 200 Increase 3.2/300 GL column (GE Healthcare) was used for increased resolution. Data was plotted in R (version 4.3.1) using the Tidyverse package [33].

## SEC-MALS

For ensemble mass determination, purified protein complex was applied to a size-exclusion column (SEC) coupled to multi angle light scattering (MALS) in DNA binding buffer (20 mM HEPES pH 7.5, 150 mM KCl, 5 mM MgCl$_2$, 3 mM BME). For each sample, 20–40 µg was added to a Superose 6 increase 10/300 column (GE Healthcare) and MALS spectra was recoded using inline DAWN-HELEOS light scattering and optilab T-rEX refractive index detectors (Wyatt). Mass was determined by analyzing the differential refractive index using the Debye model for proteins integrated in the ASTRA VI software [34]. Data was plotted in R (version 4.3.1) using the Tidyverse package [33].

## Structure determination

Xre-RES$^{Pp}$$_{Nhis6}$ and S4 dsDNA mixed in a 2:1 molar ratio with a final protein concentration of 4 mg/mL. Triagonal plate-shaped crystals were grown in drops containing 0.1M NA HEPES pH 7.5, 10% (w/v) PEG 8000, 8% ethylenglycol, and 0.2 M NaCl. All crystals were cryo-protected in reservoir solution supplemented with 20% (w/v) sucrose and flash frozen in liquid nitrogen prior to data collection. A high-quality diffraction dataset was collected at 0.937Å wavelength on a single crystal cooled to 100 K at the BioMAX beamline at MAX-IV (Lund, Sweden). The dataset was processed with XDS [35] and molecular replacement carried out using PHASER [36]. Model building was done manually in coot with iterative refinement using phenix.refine [37], DNA was built into the model gradually as density became apparent. During the final stages of refinment, phenix.refine was complemented by NAMDINATOR [38] for improved fitting and handling of rotamer outliers. The final model contains 32 chains (24 protein and 8 DNA). Xre is complete in chains B, E, F, H, I, K, N, O, Q, T, W from residues 1–149 (Xre), with chains C and U missing 149, chain R missing 147–149, and chans L and X missing 146–149. All RES proteins were missing the 13 N terminal residues from the tag, but were otherwise complete from 1-145 with the exception of chains A and J which lacks residue 145. DNA chains a and c were built from residue 8–28 (of 30 total), b and d from 3-23, e from 7-30, f from 1-24, g from 11-30, and h was from 1-20. The final structure contains, 65 water molecules.

## Mass photometry

Single molecule mass determination was carried out using the TwoMP mass photometer (Refeyn). Experimental data was acquired through one-minute videos recorded by AcquireMP (ver 2.5) software on precleaned high-precision coverslips with Culterwell gaskets (Grace Biolabs). Focus was automatically found by AcquireMP and was deemed correct when the variance was 0.01. Prior to protein analysis, a mass calibration was performed using Bovine Serum Albumin (BSA) and either Immunoglobulin M (IgM) or α2 macroglobulin (α2M) spanning a mass range of 66 kDa to 720 kDa. Standard samples were collected at final concentrations of 10 nM by 10-fold dilution into gel filtration buffer. Collected videos were analyzed in DiscoverMP (ver 2.5) by fitting Gaussian functions to ratiometric contrast values of all events in the video. Mass determination was done by fitting a linear curve between ratiometric contrast and molar mass of standard proteins. The Xre$^{Pp}$-RES$^{Pp}$ complex was diluted 10-fold, 20-fold, or 40-fold in gel filtration buffer (20 mM HEPES pH 7.5, 150 mM KCl, 5 mM MgCl$_2$, 3 mM BME) to achieve final concentrations of 10, 5, and 2 nM, respectively. Videos were collected as with the standards and mass was determined from the calibration curve. Statistical significance was defined using a Bonferroni

corrected p value of 0.0055, to avoid the multiple comparisons problem. Normality was tested using the Kolmogorov-Smirnov test, and populations were compared using the Wilcoxon test.

### Isothermal titration calorimetry

Thermodynamic constant determination was carried out using the PEAQ-ITC (Malvern panalytical). Purified protein and dsDNA samples were dialyzed into the DNA binding buffer (20 mM HEPES pH 7.5, 150 mM KCl, 5 mM $MgCl_2$, 3 mM BME), after which 20 µM of Xre-RES$^{Pp}_{Nhis6}$ was added to the sample chamber and 300–700 µM dsDNA was titrated in through 13–17 injections. Each dataset was baseline-corrected by independent measurements of buffer into buffer, titrant into buffer, and buffer into sample. Data analysis was done using Microcal PEAQ-ITC Analysis Software (Malvern panalytical). Data quality was evaluated through minimizing residuals, as well as approximating active concentrations of protein and titrant for fitting to correct N. Reported values are a mean of three independent replicates.

### Accession codes

Structure factors and coordinates for the Xre-RES$^{Pp}$:DNA complex structure have been deposited in the Protein Data Bank with PDB ID 9R35.

### Supporting information

**S1 Fig. Determination of the minimal promoter requirements of $P_{XR}$.** (A) The native DNA sequence of the $P_{XR}$ promoter. The S1, S2, S3, and S4 elements are indicated with lines and light grey boxes marks the repeat. The perfect part of the repeat in S4 is shown with a dark grey box and the mismatch in the first repeat is in red. The first 30-bp of the Xre$^{Pp}$ open reading frame (ORF) is included and marked by the bend arrow, with the N-terminal residues noted below and the start codon in bold. (B)-(F) Promoter activity assays in *E. coli* MG1655. Above, schematic representation of the promoter construct is included above each graph with DNA deletions indicated by dotted lines. Below, promoter activity measured as GFP signal in relative fluorescence units (RFU) during growth without arabinose (green curves) or with arabinose (orange curves) using the promoter reporter plasmid pGH254Kgfp with indicated promoter variations together with empty pBAD33 (-) or pBAD33::*xre-res*$^{Pp}$ (*xre-res*). $OD_{600}$ measurements for cultures without arabinose (grey curves) or with arabinose (black curves) are included. All curves represent mean-of-mean values (line, n = 3) with the standard error of the mean (SEM, shadow). The dotted vertical line indicates the 350 min time point used in Fig 2B.
(EPS)

**S2 Fig. Functional analysis of the S4 repeat sequence.** (A) Schematic representation of the $P_{XR}$ promoter. Above, the imperfect inverted repeat (red arrows with mismatch in yellow) in the S4 element is shown. Below, the sequence of the S4 highlighting the inverted repeat (light grey), the mismatch (red), and the perfect part of the repeat (dark grey). (B)-(D) Promoter activity assays in *E. coli* MG1655. Above, schematic representation of the promoter construct with DNA deletions (dotted lines) or modifications (green) indicated. Below, promoter activity measured as GFP signal in RFU during growth without arabinose (green curves) or with arabinose (orange curves) using the promoter reporter pGH254Kgfp with indicated promoter variations together with empty pBAD33 (-) or pBAD33::*xre-res*$^{Pp}$ (*xre-res*). $OD_{600}$ measurements for cultures without (grey curves) or with arabinose (black curves) are included. All curves represent mean-of-mean values (line, n = 3) with the standard error of the mean (SEM, shadow). The dotted vertical line indicates the 350 min time point used in Fig 2D.
(EPS)

**S3 Fig. Xre-RES$^{Pp}$ interacts specifically with S4 in a 1:1 protein complex to DNA duplex ratio.** (A)-(C) Analytical SEC (a-SEC) analysis of Xre-RES$^{Pp}_{NHis6}$ binding to S1 dsDNA (A), S2 dsDNA (B), or S3 dsDNA (C). Chromatograms

show elution profiles for Xre-RES$^{Pp}_{His6}$ (Protein, green), DNA (red), or DNA mixed with protein in a 1:1 ratio (Protein + DNA 1:1, purple). The full line shows 280 nm absorbance, while the dotted line is 260 nm absorbance. (D) SEC-MALS analysis of the change in mass of Xre-RES$^{Pp}_{NHis6}$ mixed with S4 dsDNA. The chromatogram shows the elution profiles for the 1:1 (blue) or the 1:2 (green) Xre-RES$^{Pp}$ complex to S4 DNA duplex ratios, as well as the masses of the peaks for the 1:1 (yellow dotted line) and 1:2 (red dotted line) ratios. The average masses corresponding to the middle of the peaks were determined to $97 \pm 3$ (1:1 ratio) and $100 \pm 8$ kDa (1:2 ratio). (E) High resolution a-SEC analysis of Xre-RES$^{Pp}_{NHis6}$ mixed with S4 dsDNA in a 1:1 (green) or 1:10 (purple) ratio. Full lines show 280 nm and dotted lines 260 nm absorbance.
(EPS)

**S4 Fig. Crystal structure of the DNA-bound Xre-RES$^{Pp}$ complex.** (A) The contents of the asymmetric unit (ASU) with electron density (2mFo-Fc, grey) contoured at 1 σ. The ASU contains four copies of the Xre-RES hexamer (mol 1-mol 4) and four copies of S4 DNA (DNA 1-DNA 4). The molecules (protein and DNA) are related through two-fold symmetry with protein in blue/green (lighter for symmetry-related molecules) and DNA in red (for DNA 1 and 3) and purple (for DNA 2 and 4). (B) A close-up view of Xre-RES molecule 1 (chains A-F) and DNA 1 (chain a) showing the high-resolution features of the map, with interacting sidechains Arg60, Arg67, Thr68, and Arg72 in sticks. (C) A focused view of the HtH domains interacting with DNA for Xre-RES (green, left) and the 434 Cro repressor from the lambda phage (PDB 1RPE) [28] (yellow, right). In each case, the structure is shown in cartoon with the HTH recognition helices colored (left) next to schematic figures showing centroid cylinder as defined by the helices (right) and including the angle between the recognition helices (green, yellow) and the DNA (faint red) (D) Promoter activity assays in *E. coli* MG1655. Activity is measured as GFP signal in RFU during growth for cultures with arabinose, using the pGH254Kgfp::P$_{XR}$ reporter plasmid combined with either empty pBAD33 (green curve), pBAD33::*xre-res*$^{Pp}$ (orange curve) or pBAD33::*xre*$^{R67A}$-*res*$^{Pp}$ (purple curve). OD$_{600}$ measurements for the pGH254Kgfp::P$_{XR}$/pBAD33 culture is included (black curve). Curves show mean-of-mean values (line, n = 3) with SEM (shadow). The dotted vertical line indicates the 350 min time point.
(EPS)

**S5 Fig. Expression of *xre*$^{Pp}$ alone only partially represses transcription from P$_{XR}$.** (A) and (B) Promoter activity assays in *E. coli* MG1655. Activity is measured as GFP signal in RFU during growth for cultures without arabinose (A) or with arabinose (B), using the pGH254Kgfp::P$_{XR}$ reporter plasmid (P$_{XR}$) combined with pBAD33::*xre* (*xre*). The colors represent six different biological replicates (n = 6), highlighting the variability in GFP expression upon *xre*$^{Pp}$ expression (B). OD$_{600}$ measurements of one of the biological replicates are included (black curve). All curves represent mean-of-mean values (line, n = 6) with SEM (shadow). The dotted vertical line indicates the 350 min time point used in Fig 5A.
(EPS)

**S6 Fig. Isolation of the 2:2 Xre-RES$^{Pp}$ complex.** (A) Analytical SEC (a-SEC) analysis. Chromatograms show elution profiles for purified the Xre-RES$^{Pp}_{His6}$ complex (4:2, green) and the Xre$_{NHis6}$-RES$^{Pp}$ complex (2:2 and free Xre antitoxin, blue). Full line shows 280 nm absorbance, while dotted line is 260 nm absorbance. The identity of peak 1 and 2 was confirmed by SDS-PAGE analysis (B).
(EPS)

**S7 Fig. AlphaFold 3 prediction of the interaction between Xre-RES and the full promoter.** (A) The AlphaFold 3-predicted structure of the hexameric Xre$_4$RES$_2$ complex bound to a 172 bp double stranded DNA segment covering to the promoter and intergenic region between *xre* and the upstream gene in the *Pseudomonas putida* KT2440 genome (P$_{XR}$), the prediction is colored by pLDDT score and regions corresponding to S3 and S4 are highlighted. (B) Predicted Aligned Error (PAE) plot corresponding to the AlphaFold 3- prediction. (C) A plot of pLDDT values as a function of residue number for the AlphaFold 3 prediction of the protein (top) and the DNA segment (bottom). (D) A comparison between the

DNA recognition site from the crystal structure (left) and the AlphaFold3 prediction (right), residues withing 3.8 Å of the DNA helix are shown in sticks and labeled.
(EPS)

**S1 Table. Promoter DNA sequences.**
(DOCX)

**S2 Table. Strains and plasmids used in this study.**
(DOCX)

**S3 Table. DNA oligonucleotides used in this study.**
(DOCX)

**S1 Text. Supporting Information Methods.**
(DOCX)

**S1 Appendix. Source Data for Figures and Tables.**
(ZIP)

## Acknowledgments

The authors would like to thank Janni Nielsen and Prof. Daniel E. Otzen for help with SEC-MALS, Camilla G. Andersen for help with XRD data analysis, Rasmus Freund for scripts designed to plot the pLDDT and PAE files from the AlphaFold 3 webserver, and Dr. Kristoffer Winther for help with identification of the four promoter repeat sequences. We also wish to thank Associate professor Magnus Kjærgaard and the MBG Biophysics Core Facility for use of equipment supported by The Carlsberg Foundation. We acknowledge MAX IV Laboratory for time on Beamline BioMAX under Proposal 20220062. Research conducted at MAX IV, a Swedish national user facility, is supported by the Swedish Research council under contract 2018–07152, the Swedish Governmental Agency for Innovation Systems under contract 2018–04969, and Formas under contract 2019–02496.

## Author contributions

**Conceptualization:** Frederik Oskar Graversgaard Henriksen, Ditlev Egeskov Brodersen, Ragnhild Bager Skjerning.

**Data curation:** Ragnhild Bager Skjerning.

**Formal analysis:** Frederik Oskar Graversgaard Henriksen, Ragnhild Bager Skjerning.

**Funding acquisition:** Ditlev Egeskov Brodersen, Ragnhild Bager Skjerning.

**Investigation:** Frederik Oskar Graversgaard Henriksen, Lan Bich Van, Ragnhild Bager Skjerning.

**Methodology:** Frederik Oskar Graversgaard Henriksen, Lan Bich Van, Ragnhild Bager Skjerning.

**Project administration:** Ragnhild Bager Skjerning.

**Supervision:** Ditlev Egeskov Brodersen, Ragnhild Bager Skjerning.

**Validation:** Ragnhild Bager Skjerning.

**Visualization:** Frederik Oskar Graversgaard Henriksen, Ditlev Egeskov Brodersen, Ragnhild Bager Skjerning.

**Writing – original draft:** Frederik Oskar Graversgaard Henriksen, Ditlev Egeskov Brodersen, Ragnhild Bager Skjerning.

**Writing – review & editing:** Frederik Oskar Graversgaard Henriksen, Lan Bich Van, Ditlev Egeskov Brodersen, Ragnhild Bager Skjerning.

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
