## [Decision Letter · Decision Letter 0]

PGENETICS-D-25-00196

Structural basis for higher-order DNA binding by a bacterial transcriptional regulator

PLOS Genetics

Dear Dr. Bager Skjerning,

Thank you for submitting your manuscript to PLOS Genetics. After careful consideration, we feel that it has merit but does not fully meet PLOS Genetics's publication criteria as it currently stands. Therefore, we invite you to submit a revised version of the manuscript that addresses the points raised during the review process.

Please submit your revised manuscript within 60 days May 25 2025 11:59PM. If you will need more time than this to complete your revisions, please reply to this message or contact the journal office at plosgenetics@plos.org. Please include the following items when submitting your revised manuscript:

We look forward to receiving your revised manuscript.

Kind regards,

Kai Papenfort

Academic Editor

PLOS Genetics

Danielle Garsin

Section Editor

PLOS Genetics

Aimée Dudley

Editor-in-Chief

PLOS Genetics

Anne Goriely

Editor-in-Chief

PLOS Genetics

**Additional Editor Comments:**

Dear Dr. Bager Skjerning.

Thank you for submitting your work to PLOS Genetics. The manuscript has now been reviewed by three experts in the field and their comments are provided below. While the referees find your work potentially interesting, they also raised substantial criticisms. We will consider publishing your manuscript only if you can address these criticisms in a revised version of the manuscript. Of note, one referee indicated that the PDB deposition code and validation summary are missing. Please make sure this is addressed in the revised manuscript.

Kind regards,

Kai Papenfort

**Journal Requirements:**

At this stage, the following Authors/Authors require contributions: Frederik Oskar Graversgaard Henriksen, Lan Bich Van, and Ditlev Egeskov Brodersen. Please ensure that the full contributions of each author are acknowledged in the "Add/Edit/Remove Authors" section of our submission form.

The list of CRediT author contributions may be found here: https://journals.plos.org/plosgenetics/s/authorship#loc-author-contributions

https://journals.plos.org/plosgenetics/s/submission-guidelines#loc-parts-of-a-submission

4) We do not publish any copyright or trademark symbols that usually accompany proprietary names, eg ©,  ®, or TM  (e.g. next to drug or reagent names). Therefore please remove all instances of trademark/copyright symbols throughout the text, including:

- TM on page: 20 and 23.

5) Please upload all main figures as separate Figure files in .tif or .eps format. For more information about how to convert and format your figure files please see our guidelines: 

6) We notice that your supplementary Figures, and Tables are included in the manuscript file. Please remove them and upload them with the file type 'Supporting Information'. Please ensure that each Supporting Information file has a legend listed in the manuscript after the references list.

7) Please ensure that the funders and grant numbers match between the Financial Disclosure field and the Funding Information tab in your submission form. Note that the funders must be provided in the same order in both places as well.

**Reviewers' comments:**

Reviewer's Responses to Questions

**Comments to the Authors:**

Reviewer #1: The manuscript from Henriksen et al describes microbiological, biochemical and structural analysis of transcriptional regulation for the Xre-RES toxin antitoxin module from Pseudomonas putida. Whilst having the hallmarks of standard TA systems and transcriptional regulators - TA complexes acting together to negatively autoregulate, use of HTH domains, promoters with palindromic repeats and tandem repeats - the observed phenotypes demonstrate nuances to the regulatory process. These findings demonstrate plasticity in the TA complex itself, use of repeats in the promoter region and also in protein-DNA binding interfaces.

The manuscript was a pleasure to read. Very clear and logical. The experiments are laid out clearly and easy to follow. The conclusions are supported by the presented data and the model looks fair to me. This is a lovely addition to a field that is increasingly showing a depth of diversity and individual specificities to govern biological behaviour.

I have some suggestions:

Line 119 - the bacteria are transformed, not the constructs.

Line 123 - is there a sentence or two missing? The data in 1D are not discussed (though they clearly show repression)

Line 148 - "of" missing from the end

Line 155 - "the" missing

Line 183 - remove "an"

Fig 4F - The arrangement of the 2x DNAs suggests another go at your SEC and ITC experiments, using an extended DNA probe that goes beyond the 30 bp already used, would likely provide data to support a model of perhaps tighter binding through engagement with a second site. Your discussion implies perhaps you are saving this for the next manuscript so I don't think the experiment should be a requirement here, merely a possible addition.

Table 1 - Please add in ramachandran allowed/outliers etc values

Reviewer #2: Frederik Oskar Graversgaard Henriksen and colleagues provide a solid structural and functional characterization of Xre-RES binding to its own promoter to repress transcription. Their study introduces a concentration-dependent model of transcriptional autoregulation by the TA system, supported by X-ray crystallography, size-exclusion chromatography (SEC), AlphaFold3 predictions, and in vivo promoter activity assays.

While the study is well-executed, I have several concerns and suggestions:

1-Promoter activity assays: The authors analyze promoter activity using in vivo GFP assays but do not perform β-galactosidase assays in E. coli, which are a standard approach for measuring transcriptional activity. Was this method considered? Additionally, could the authors clarify why the RFU curves show a decline from initially strong fluorescence intensity? Given the relevance of Pseudomonas putida to the native system, would promoter activity assays in this organism provide additional insights? To complement the fluorescence-based promoter activity assays, a Western blot analysis of GFP protein levels would be beneficial.

2-Direct interaction between Xre-RES and the promoter: While structural data provide strong support for DNA binding, traditional biochemical validation, such as electrophoretic mobility shift assays (EMSA), could further confirm the interaction. Have the authors performed EMSA or similar experiments?

3-In Figure 4E, the annotation and functional validation: There may be an annotation issue in Figure 4E. Could the authors clarify this? Furthermore, an in vitro mutagenesis assay to assess the essentiality of residue R67 in Xre-RES for the DNA binding activity would strengthen their conclusions.

4-In Figure 5, when presenting AlphaFold3 predictions, it would be helpful to include confidence metrics such as PTM, iPTM scores. Additionally, a more detailed analysis of predicted interaction interfaces (e.g., hydrogen bonding networks) would improve the interpretation of the structural model.

5-Figure resolution: several figures appear to have low resolution, making it difficult to assess structural details. Higher-quality images would enhance clarity, particularly for structural and interaction data.

Minor points:

-Figure 1D is not described or interpreted in the manuscript. Could the authors clarify its significance?

-In figure S5, with the addition of arabinose, the cells appear to grow better. Could the authors explain this phenomenon?

Overall, this study provides valuable mechanistic insights into the transcriptional autoregulation of Xre-RES. Addressing these points would further strengthen the manuscript.

Reviewer #3: In this manuscript authors study structural basis of DNA binding by a Xre-RES complex. They find that the antitoxin-toxin complex transitions between 2:2 to 4:2 ratios depending on the concentration and that only the 4:2 ratio complex binds the promoter region to repress the transcription of the xre-res genes. In vitro Xre-RES complex binds specifically to S4 element and in vivo likely to a longer region including S3 element. They also conclude that Xre alone is monomeric and highly unstable. RES is therefore required for formation of Xre dimers that can bind DNA. Manuscript is overall well written, however apart from a few novel details, the conclusions are rather expected and similar to those that have been reached for other homologous Xre-RES TAs from other bacteria. Of note, this model of regulation and transition between different order and ratio complexes is conserved for even structurally unrelated TAs, and this is missing in the discussion.

Major remarks:

Line 122 – statement that PXR promoter becomes increasingly active during exponential growth can be questioned – authors used stable GFP protein and the increased signal could be due to progressive accumulation of synthesized GFP protein.

In figure 2 the repeats (operator sites) are discussed and sampled extensively, but there’s no information about the actual promoter – where is it located? Does it overlap with one or few proposed regulation elements? Has it been verified experimentally, for example by 5’ RACE?

In figure 2 panel B and D, the first graph with native PXR promoter seems to be the same data based on data points and heights of the bar graphs – if the same data has been plotted (all the experiments performed at the same time?), it should probably be mentioned. Why do the p values differ for these two seemingly identical pairs of bar graphs?

Figure 4 – it seems like the DNA used for crystallization was not sufficiently long and therefore Xre-RES complex interacts with a DNA from a symmetry-related complex. As authors conclude – only one Xre dimer of two in the complex covers the S4 site, and secondary site inside full operator shall exist. However, the interpretation of the crystallized complex is then difficult, since the discussed distances would likely change in native context. This is partially addressed in comparison to the AF-predicted structure, but authors interpret it as AF-mistake.

Discussion also lacks insights on the dimerization of Xre and the reason for its instability as the monomer. It is likely, that each monomer has different interactions with the toxin, this would bring them close together composing a DNA-binding dimer – this is a bit unusual as compared to stronger antitoxin dimers, stabilized by strand exchange for example in cases of RHH- DNA binding domain. However, in contrast to other TA-antitoxins Xre does not seem to have intrinsically disordered regions, and yet curiously it is unstable without its toxin. Are there biochemical or physiological reasons for its instability?

Minor:

Line 203 – incomplete phrase- did authors mean “at different protein:DNA ratios”?

**Have all data underlying the figures and results presented in the manuscript been provided?**

Reviewer #1: Yes

Reviewer #2: None

Reviewer #3: **No: ** PDB deposition code and validation summary is missing

PLOS authors have the option to publish the peer review history of their article (what does this mean? ). If published, this will include your full peer review and any attached files.

**Do you want your identity to be public for this peer review?** For information about this choice, including consent withdrawal, please see our Privacy Policy .

Reviewer #1: No

Reviewer #2: No

Reviewer #3: No

**Figure resubmission:**
---

## [Decision Letter · Decision Letter 1]

PGENETICS-D-25-00196R1

Structural basis for higher-order DNA binding by a bacterial transcriptional regulator

PLOS Genetics

Dear Dr. Bager Skjerning,

Thank you for submitting your manuscript to PLOS Genetics. After careful consideration, we feel that it has merit but does not fully meet PLOS Genetics's publication criteria as it currently stands. Therefore, we invite you to submit a revised version of the manuscript that addresses the points raised during the review process.

Please submit your revised manuscript within 30 days Jun 24 2025 11:59PM. If you will need more time than this to complete your revisions, please reply to this message or contact the journal office at plosgenetics@plos.org. Please include the following items when submitting your revised manuscript:

We look forward to receiving your revised manuscript.

Kind regards,

Kai Papenfort

Academic Editor

PLOS Genetics

Danielle Garsin

Section Editor

PLOS Genetics

Aimée Dudley

Editor-in-Chief

PLOS Genetics

Anne Goriely

Editor-in-Chief

PLOS Genetics

**Additional Editor Comments :**

Dear Dr. Bager Skjerning.

I am glad to say that the reviewers have commented positively on your revised paper and your manuscript can now be considered as ‘accepted in principle’. However, before I formally accept the paper, please take a close look at the comment provided by reviewer #1. S/he raises an important point regarding Table 1 that I think should be considered.

Best wishes and congratulations on a very nice manuscript,

Kai Papenfort

(Academic Editor)

**Reviewers' comments:**

Reviewer's Responses to Questions

Reviewer #1: The revisions to the manuscript by Henriksen et al look good except on one point. In my initial review I'd asked the authors to put Ramachandran values into Table 1. The rebuttal says this has been done, but looking at Table 1 in the revision the values are still missing - unless I am missing something? 0.92% Rama outliers (as I can see from edits in MandMs in the marked version) is also a little high. Can you that down any further? Otherwise, looks all fine to me.

Congrats to the authors.

Reviewer #2: Thank you for the authors' reply, which has addressed all of my concerns.

Reviewer #3: Authors have addressed all the comments of the reviewers, but the increment to the quality of the manuscript is minimal, since no additional experimental support was provided.

**Have all data underlying the figures and results presented in the manuscript been provided?**

Reviewer #1: Yes

Reviewer #2: Yes

Reviewer #3: Yes

PLOS authors have the option to publish the peer review history of their article (what does this mean? ). If published, this will include your full peer review and any attached files.

**Do you want your identity to be public for this peer review?** For information about this choice, including consent withdrawal, please see our Privacy Policy .

Reviewer #1: No

Reviewer #2: **Yes: ** Xibing Xu

Reviewer #3: No

**Figure resubmission:**
---

## [Editor Report · Decision Letter 2]

Dear Dr Bager Skjerning,

We are pleased to inform you that your manuscript entitled "Structural basis for higher-order DNA binding by a bacterial transcriptional regulator" has been editorially accepted for publication in PLOS Genetics. Congratulations!

Yours sincerely,

Kai Papenfort

Academic Editor

PLOS Genetics

Danielle Garsin

Section Editor

PLOS Genetics

Aimée Dudley

Editor-in-Chief

PLOS Genetics

Anne Goriely

Editor-in-Chief

PLOS Genetics

Comments from the reviewers (if applicable):

Congrats on a very nice manuscript.

Best wishes

Kai Papenfort

**Data Deposition**

http://datadryad.org/submit?journalID=pgenetics&manu=PGENETICS-D-25-00196R2

**Press Queries**

---

## [Editor Report · Acceptance letter]

PGENETICS-D-25-00196R2

Structural basis for higher-order DNA binding by a bacterial transcriptional regulator

Dear Dr Bager Skjerning,

We are pleased to inform you that your manuscript entitled "Structural basis for higher-order DNA binding by a bacterial transcriptional regulator" has been formally accepted for publication in PLOS Genetics! Your manuscript is now with our production department and you will be notified of the publication date in due course.

With kind regards,

Zsofia Freund

PLOS Genetics

On behalf of:
